# Use of Bladder-Related Medication in Non-Muscle Invasive Bladder Cancer Patients

**DOI:** 10.3390/cancers16101936

**Published:** 2024-05-20

**Authors:** Linea Blichert-Refsgaard, Charlotte Graugaard-Jensen, Mette Nørgaard, Jørgen Bjerggaard Jensen

**Affiliations:** 1Department of Urology, Aarhus University Hospital, Palle Juul-Jensens Boulevard 35, 8200 Aarhus N, Denmark; charlotte.graugaard-jensen@rm.dk (C.G.-J.); bjerggaard@skejby.rm.dk (J.B.J.); 2Department of Clinical Epidemiology, Aarhus University Hospital, Olof Palmes Alle 43-45, 8200 Aarhus N, Denmark; mn@clin.au.dk

**Keywords:** non-muscle invasive bladder cancer (NMIBC), anticholinergics, β3-agonists, cystitis-relevant antibiotics, transurethral bladder resections (TURBs), instillation treatment

## Abstract

**Simple Summary:**

Treatment of non-muscle invasive bladder cancer (NMIBC) consists mainly of repeated transurethral bladder surgery (TURBs) and instillation treatments. This treatment might influence bladder function and, therefore, the patient’s quality of life. We used bladder-related medication as a surrogate marker of compromised bladder function to investigate whether the load of repeated surgery and the exposure to adjuvant instillation influence bladder function. The study was conducted as a register-based study in a non-selected national cohort of NMIBC patients. We included 17,774 patients. Patients exposed to ≥5 TURBs had a higher risk of using bladder-relaxing medication than patients exposed to 1 TURB and a higher risk of cystitis. BCG-exposed patients had a higher risk of bladder-relaxing medication use compared to non-exposed and had a higher risk of cystitis. Repeated TURBs have the highest impact on bladder function. To maintain the best possible bladder function, the goal of NMIBC treatment should be as few TURBs as feasible.

**Abstract:**

Repeated transurethral bladder resections (TURBs) and instillation treatments in non-muscle invasive bladder cancer (NMIBC) might influence bladder function and, therefore, quality of life. Bladder-related medication is a surrogate marker of compromised bladder function. The objective was to investigate whether TURBs and adjuvant instillation therapy are associated with the use of anticholinergics, β3-agonists, and cystitis-relevant antibiotics. We divided all Danish patients diagnosed with primary NMIBC during 2002–2017 registered in the Danish National Patient Registry (DNPR) based on TURB-load within the first five years from diagnosis (1 TURB, 2–4 TURBs, ≥5 TURBs). Instillation therapy with either mitomycin C (MMC) or bacillus Calmette-Guerin vaccine (BCG) was independent exposure (yes or no). We included 17,774 patients; 76% men, median age: 70 years (IQR: 63, 77). Patients exposed to ≥5 TURBs had a higher risk of using bladder-relaxing medication than patients exposed to 1 TURB, HR = 4.01 [3.33; 4.83], and higher risk of cystitis, HR = 2.27 [2.05; 2.51]. BCG-exposed patients had a higher risk of bladder-relaxing medication use compared to non-exposed, HR = 1.92 [1.69; 2.18], and a higher risk of cystitis, HR = 1.39 [1.31; 1.48]. Repeated TURBs have the highest impact on bladder function. Adjuvant instillation therapy is also associated with the use of bladder-related medication.

## 1. Introduction

Bladder cancer is a commonly diagnosed cancer worldwide, with the highest incidence rates in developed countries and almost three times higher incidence in men than in women [1]. In Denmark, nearly 2000 new patients are diagnosed with bladder cancer each year [2]. Approximately 75% of patients with bladder cancer present with non-muscle invasive bladder cancers (NMIBCs).

NMIBC is characterized by a high risk of recurrence and a risk of progression to muscle-invasive bladder cancer; clinical studies have shown an estimated five-year risk of recurrence of NMIBC of up to 78% and a five-year risk of progression of up to 45% [3].

Transurethral resection of the bladder (TURB) is the primary diagnostic treatment for NMIBC. Depending on the histopathological stage and grade of the bladder tumor, recurrence-preventing adjuvant intravesical instillation therapy with mitomycin C (MMC) or bacillus Calmette-Guerin vaccine (BCG) is recommended in the European Association of Urology (EAU) guidelines, as well as the Danish guidelines [4,5].

The high recurrence risk results in a high risk of repeated TURBs. Several clinical studies have shown the great clinical importance of TURB quality with regard to the prognosis of NMIBC, and the prevalence of detrusor muscle in the specimen is considered a surrogate of high-quality TURB because of a more accurate diagnostic and staging [6,7,8]. However, repeated deep resections will leave fibrotic scars in the bladder wall and potentially cause reduced compliance and bladder shrinkage, which will disturb normal bladder function. Thus, a late effect of repeated TURBs could be frequency, urgency, and incontinence similar to the overactive-bladder syndrome [9].

Irritative bladder symptoms are known side effects of adjuvant instillation therapy, especially BCG [10,11]. Frequent involuntary contractions of the detrusor muscle may occur because of immune-modulated irritation of the bladder wall induced by instillation treatment.

A cornerstone in the first-line treatment of overactive bladder, urgency, and urge incontinence is centered on the control of detrusor contractions with anticholinergics or β3-agonists (Mirabegron).

Cystitis is a possible complication to every treatment in the urinary tract, including TURB and instillation treatment. Clinically significant cystitis is treated with targeted antibiotics; in Denmark, the most commonly used antibiotics in cystitis treatment are pivmecillinam, trimethoprim, sulfamethiziole, nitrofurantoin, ciprofloxacin, and amoxicillin + clavulanic acid.

In Denmark, information in the Danish nationwide registries can be linked on an individual level through the unique 10-digit Civil Personal Register (CPR) number, a type of social security number assigned to all Danes at birth [12]. Therefore, the Danish national registries provide a unique opportunity to study bladder-related medicine use in an everyday setting correlated to treatment in a non-selected NMIBC patient cohort.

We aimed to assess the use of anticholinergics, β3-agonists, and cystitis-relevant antibiotics in NMIBC patients dependent on TURB-load and the exposure to adjuvant instillation therapy through a nationwide, population-based study. Using the prescriptions of these drugs as a surrogate marker of compromised bladder function, we aimed to assess a possible treatment-dependent difference to ensure the most bladder-beneficial NMIBC treatment in the future.

## 2. Materials and Methods

### 2.1. Setting

We conducted a nationwide cohort study. The Danish population was approximately 5.7 million people in 2017 [13]. The Danish healthcare system provides free, tax-supported medical services to all Danes. Created in 1977 and including all information on diagnoses and treatments, the Danish National Patient Registry (DNPR) is internationally considered one of the most comprehensive of its kind [14]. The validity of the coding in the DNPR is generally found to be very high, which also applies to the codes related to NMIBC treatment [15,16]. Since 1994, the International Classification of Diseases, 10th Edition (ICD-10) has been used to classify the diagnoses and procedures in the DNPR.

In Denmark, only very few patients are treated for NMIBC in a private setting, with the vast majority being treated at public hospitals. Since 2003, private hospitals have likewise registered procedures and diagnoses in DNPR but the registration is known to be less complete [14].

The Danish Pathology Registry (DPR) has been nationwide since 1997 and contains all pathological diagnoses in SNOMED codes, which includes information on histological subtype, tumor grade, and tumor stage [17,18].

In Denmark, neoplastic bladder lesions are classified using the TNM classification, and pathological graded by WHOs 2004 classification system, before 2009 Bergkvist classification was used in Denmark [19,20,21].

The National Prescription Registry (NPR) includes data on all drugs sold in primary care or purchased for use in Danish hospitals. The NPR data follow the Anatomical Therapeutic Chemical (ATC) classification system (described in full elsewhere [22]). Danish pharmacies receive a financial incentive for complete registration of all purchases, and the electronic, code-based prescription system and dispensing process at Danish pharmacies minimize the risk of data entry errors. The NPR is considered both complete and valid as of 1995 [23]. Drugs prescribed to nursing home residents are also included. Drug information not included in NPR, on an individual level, is information on drugs used during hospital admissions, over-the-counter drugs, drugs used by certain institutionalized individuals (typically due to psychiatric illnesses), and drugs supplied directly by hospitals or treatment centers (e.g., chemotherapeutic agents) [23].

Because of the unique CPR number, all diagnoses and treatments from the DNPR can be linked to, e.g., a pathological diagnosis from the DPR and prescription information from the NPR on an individual and time-specific level [24].

### 2.2. Study Population

Through the DNPR, we identified all Danish patients diagnosed with primary bladder cancer from 2002 to 2017. Inclusion required the following diagnoses: bladder cancer, the pathological diagnosis of neoplastic bladder lesion, and one of the following procedures as the first registered procedure: cystoscopy alone, cystoscopy with a biopsy, or TURB. The codebook shows all codes used in the sampling (Appendix A).

To avoid misclassification of cancer treatment for other cancer types as bladder cancer treatment, we excluded all patients with a cancer diagnosis other than bladder cancer, prostate cancer, and non-melanoma skin cancer. Furthermore, we excluded all patients treated at private hospitals.

Patients with suspicion of muscle-invasive bladder cancer (TNM ≥ T2 and classification of “at least pT1”) at the time of initial bladder cancer diagnosis and patients undergoing cystectomy upfront were excluded.

### 2.3. Definitions

We defined a TURB as an additional pathological bladder cancer diagnosis registered in the DPR.

We defined drug use as a collection of ≥2 prescriptions filled of a drug of interest and non-use as 0–1 prescriptions filled, applicable to both bladder-relaxing agents (anticholinergics and β3-agonists) and cystitis-relevant antibiotics.

### 2.4. Statistical Analysis

Using the TURB load within the first five years from diagnosis as exposure, we divided the study cohort into three groups: Group 1: one TURB (minimal surgical load), Group 2: 2 to 4 TURBs (intermediate surgical load), and Group 3: five or more TURBs (significant surgical load).

Using exposure to adjuvant instillation therapy with either BCG or MMC within the first five years from NMIBC diagnosis as independent exposures, we divided the study cohort into two exposure groups: exposure to adjuvant installation therapy (Group 1) and no exposure (Group 2). We analyzed BCG and MMC treatment separately.

Our outcome was time-to-event, and we considered death and cystectomy as competing risks.

To minimize bias by misclassification of the categorical exposure, we allowed patients to enter the exposure groups with delayed entry, resulting in dependent exposure groups. Therefore, every patient could contribute to more than one group if they reached the next level of exposure before reaching the event of interest or one of the competing risks within five years of NMIBC diagnosis. Under the proportional hazards assumption, we used Cox regression analyses resulting in estimates of hazard ratios (HRs) using the exposure groups 1 (only one TURB, no exposure to MMC, and no exposure to BCG, respectively) as reference. All estimates are presented with 95% confidence intervals.

We additionally adjusted the results in multivariate analyses, including adjuvant instillation with BCG (yes or no) and adjuvant instillation with MMC (yes or no) together with the exposure by TURB-load.

We used Aalen–Johansen curves in the graphic presentation of the cumulative incidences of bladder-relaxing agent use and use of cystitis-relevant antibiotics. The Aalen–Johansen curve is suitable for competing risks and delayed entry in exposure use. To make the comparison within the TURB-load exposure groups reasonable, a one-year lag time was added to these curves simply because it takes time to reach five or more TURBs.

We furthermore stratified the study cohort on exposure to adjuvant instillation therapy within the first five years from NMIBC diagnosis. We did sub-analyses on patients exposed to BCG or MMC, respectively. We anticipated that a higher number of instillations would result in a higher rate of bladder-related medication collection, and the sub-analyses, therefore, aimed to assess the influence of accumulated instillations on bladder function. We defined a significant burden of instillations as more than six instillations within the first two years from diagnosis corresponding to the initiation of either maintenance or re-induction treatment. Resulting in two groups: low accumulated instillations and high accumulated instillations. Because of the fixed exposure time (more than six instillations at two years: yes or no), we applied a two-year lag time to these sub-analyses to minimize the risk of immortal-time bias.

We used STATA Statistical Software: Release 17, StataCorp LLC 905 Lakeway Drive College Station, TX 77845-4512, USA in all data management, analysis, and graphics manufacturing.

## 3. Results

We identified 17,774 patients diagnosed with primary NMIBC in Denmark between 2002 and 2017 meeting the inclusion criteria (Figure 1). Of these, 76% were men, and the median age at diagnosis was 70 years (IQR: 63, 77). Nearly 7% and 51% of the patients in the entire cohort had collected prescriptions of bladder-relaxing agents or cystitis-relevant antibiotics, respectively, before NMIBC diagnosis (Table 1).

Primary histology is slightly unequally distributed. TURB-group 3 (≥five TURBs) had a higher proportion of low-risk NMIBC compared to TURB-group 1 (Appendix A), and the groups exposed to instillation treatments, BCG and MMC, had a lower proportion of low-risk patients, and a correspondingly larger proportion of high-risk NMIBC patients, than the groups not exposed to instillation treatments (Appendix A).

Compared to patients with a minimal burden of TURBs, patients with a severe surgical burden of TURBs within the first five years from NMIBC diagnosis had a higher risk of collecting bladder-relaxing agents, anticholinergics and β3-agonists, with an HR of 4.01 95% confidence interval [3.33; 4.83] (Table 2). Patients with exposure to BCG instillations had a higher risk of collecting prescriptions on bladder-relaxing medication compared to patients without BCG exposure; HR was 1.92 [1.69; 2.18]. The HRs only attenuated slightly after adjustment (Table 2).

Patients with a significant surgical burden of TURBs had a higher risk of cystitis compared to patients with a minimal surgical burden, HR = 2.27 [2.05; 2.51], and patients exposed to BCG installation had a higher risk of cystitis than patients without—HR = 1.39 [1.31; 1.48]—adjustments did not remove this association (Table 2).

Figure 2 and Figure 3 show the Aalen–Johansen curves of the non-adjusted cumulated incidence estimates of bladder-relaxing agents-use and use of cystitis-relevant antibiotics, individual curves representing exposure of TURB-load (a) and BCG instillations (b). The ten-year cumulated incidence of collecting two or more prescriptions of bladder-relaxing agents when exposed to a minimal surgical load of only one TURB was approximately 5%, and when exposed to five or more TURBs within the first five years from NMIBC diagnosis, this cumulated incidence was approximately 13%. With regard to the collection of more than two prescriptions of cystitis-relevant antibiotics, the ten-year cumulated incidence in TURB-exposure group 1 was approximately 50%, compared to approximately 60% in TURB-exposure group 3. The 10-year cumulated incidence of collecting two or more prescriptions of bladder-relaxing agents with no exposure to adjuvant BCG treatment was approximately 10%, and when exposed to BCG instillations within the first five years from NMIBC diagnosis, the cumulated incidence was approximately 14%. With regard to the collection of more than two prescriptions of cystitis-relevant antibiotics, the ten-year cumulated incidence in the group not exposed to BCG was approximately 61%, compared to approximately 62% in the group exposed to BCG. The MMC-exposed part of the study cohort was proportionally too small for a sensible graphic representation of the cumulative incidences of the events of interest.

### Sub Analyses: BCG-Treated Patients and MMC-Treated Patients

Table 3 shows the HRs on collecting two or more prescriptions of bladder-relaxing agents (anticholinergics and β3-agonists) and cystitis-relevant antibiotics depending on the amount of BCG or MMC instillations in the stratified cohort of BGC- or MMC-treated NMIBC-patients, respectively. The HR of bladder-relaxing agent use between patients exposed to more than six BCG instillations within the first two years from NMIBC diagnosis (high accumulated instillations) and the patients exposed to six or less BCG instillations (low accumulated instillations) was 1.34 [0.97; 1.86]. The HR of cystitis-relevant antibiotic use between the groups was 0.80 [0.63; 1.02]. The HR of bladder-relaxing agent use between patients exposed to more than six MMC instillations (high accumulated instillations) and the patients exposed to six or less MMC instillations (low accumulated instillations) was 0.52 [0.21; 1.30]. The HR of cystitis-relevant antibiotic use between the groups was 0.95 [0.54; 1.66].

## 4. Discussion

In this study, we found a clear association between the surgical burden following TURBs within the first five years since NMIBC diagnosis and collected prescriptions of anticholinergics, β3-agonists, and cystitis-relevant antibiotics. Increasing amounts of TURBs result in more use of bladder-related medicine. We also found that the use of instillation therapies within the first five years of NMIBC diagnosis was associated with the collection of anticholinergics, β3-agonists, and cystitis-relevant antibiotics.

Surprisingly, the number of bladder installations within the first two years from diagnosis was not clearly associated with the use of anticholinergics, β3-agonists, and cystitis-relevant antibiotics. Instillation therapies are known to be important in NMIBC recurrence-reduction, but it is also associated with the development of bladder-related adverse events and impacts QOL, especially during BCG treatment [10,25,26]. In our study, we found a tendency towards an association between the accumulated amounts of BCG instillations and the use of bladder-relaxing agents. We expected a more significant association between both MMC and BCG instillations and the collection of bladder-related-medicine prescriptions in general. An explanation could be a possible misclassification bias followed by the fixed-time exposure groups in the sub-analysis and the fact that we introduced a time lag of two years. Consequently, patients receiving installation number seven after this timeframe would be categorized as having low accumulated instillations, resulting in a bias of the estimate of the association between accumulated amounts of instillations and bladder impact towards zero. Additionally, the two-year lag time may minimize the immortal time bias but may instead introduce a selection bias due to the no-censoring requirement within the first two years from NMIBC diagnosis, meaning that the patient needs to survive the first two years without cystectomy and an event of interest (collection of two or more prescriptions of bladder-relaxing agents or cystitis-relevant antibiotics). This will, most likely, also lead the estimated association towards zero since the patients with an early, large impact on urinary function due to early severe accumulated instillations would be excluded from the sub-analyses.

The unequal primary-histology distribution across exposure groups of adjuvant instillation therapy is expected according to guideline-recommended risk stratification, especially regarding BCG (Appendix A) [5]. Likewise, the very small percentage of primary pT1b in the exposure group with the largest TURB load is also expected as pT1b, according to Danish guidelines, should be considered for cystectomy and will, therefore, rarely reach five TURBs (Appendix A). If low/intermediate risk at primary histology was an independent factor for collecting two or more prescriptions of bladder-related medicine, it could potentially lead to a bias of the estimated association between exposure, TURB-load or instillation therapy, and the outcome, bladder-related medicine prescription collection. This could hypothetically result in an overestimation of the association between TURB-load and medicine collection and an underestimation of the association between instillation therapy and bladder-related medicine collection. However, we have no reason to believe this to be the case, and furthermore, looking specifically at the distribution of pTa LG in the TURB-exposure groups, it is smaller in TURB-group 2 (2–4 TURBs) than in TURB-group 1 (one TURB), but the cumulated HRs still shows a strong association between TURB-load and collection of prescriptions for bladder-related medicine.

We know from other studies that adherence to guidelines regarding instillation therapies in NMIBC treatment is low [27,28]. This corresponds to our clinical experience in Denmark. The Danish guidelines have a less powerful recommendation of MMC instillations than the EAU guidelines, probably resulting in very low use of MMC in primary NMIBC treatment, where most instillations are initiated following recurrences [4,5]. Thus, the lag time of two years might be too short to ensure enrolment of a representative high accumulated instillations cohort. We chose the two years as we wanted the time slot short enough to ensure a significant accumulation of instillations over time and because we wanted the clearest association between instillation and bladder function impact. Allowing a larger time slot would blur the picture because of more TURBs and would enhance selection bias.

Furthermore, a patient with impaired bladder function already at induction treatment will never receive re-induction or maintenance, also resulting in a bias of the associated estimate towards the null as they would never reach instillation number seven. 

Regarding the use of cystitis-relevant antibiotics, where a proportion of the larger use correlated to a higher number of TURBs, this could be caused by the preventive use of antibiotics during surgery. As such, the collection of cystitis-relevant antibiotics might not be a one-to-one surrogate of clinically important cystitis symptoms and bladder impact. Nevertheless, a larger use of antibiotics, preventive or not, contributes to the accumulated risk of developing antibiotic resistance [29], and every attempt to minimize antibiotic use should be coveted.

Our study did not include a control group without NMIBC. This compromises the external validity of the results, but the aim of our study was to assess the association between a minimal, an intermediate, and a significant surgical burden of TURBs and the following impact on the bladder function and, additionally, the bladder-function-consequences following instillation therapy for NMIBC patients.

The recurrence rate, and thus the number of TURBs, as well as the exposure to instillation therapy, depends, to some extent, on the histology of the removed bladder tumor. Due to design, it was not possible to adjust for the histology prior to each individual TURB or instillation treatment, but independent of the histology, the aim of our study was to assess the bladder function-related consequences of the NMIBC treatment.

Further limitations are mainly due to design and the limited information in the registries. It has previously been speculated that patients with larger tumors and/or tumors located in the trigone or bladder neck might have a more pronounced change in urinary function after treatment [30], but the DNPR does not contain information on the size and location of tumors. This introduces a random error, but we expect this to be eliminated due to the very large size of our study population. Additionally, one major limitation of the NPR is the lack of information on the indication of drug prescription/use, intended duration, and dosage. However, we do not expect this bias to differ between the exposure groups in either main- or sub-analyses, and therefore, it does not influence the estimated hazard ratios.

Health-related quality of life (HRQOL) among bladder cancer survivors is worse than among survivors of other pelvic cancers [31], and the compromised bladder function with involuntary detrusor contractions leading to frequency, urgency, incontinence, and cystitis issues, could be a part of the explanation. Undergoing instillation therapy, in itself, is known to affect HRQOL negatively [32]. Garg et al. did a qualitative study assessing urinary function as an important factor for patients and a priority issue for the caregiver in improving HRQOL in NMIBC treatment [33]. Other studies have also described the relationship between compromised bladder function and a lesser QOL score, especially due to storage capacity and the social functioning domains correlated herewith [25,34]. In our study, we examined the collection of two or more prescriptions for anticholinergics, β3-agonists, and cystitis-relevant antibiotics, resulting in ignorance of all events where only one prescription was collected. González-Padilla et al. did a prospective observational study of 56 patients receiving induction-course bladder instillations of BCG, MMC, or chemohyperthermia (CHT) with MMC; 7% of BCG and CHT MMC-treated patients did not complete the induction course due to adverse events [34]. Nevertheless, they found no clinically significant differences between the groups regarding QOL and lower urinary tract symptoms during the induction course, and all patients recovered to their baseline QOL at the end of induction. This could support our contention that the side effects of instillation therapy are more acute and reversible, and, as such, we did not see the impact as strongly when concentrating the outcome on the collection of more than one prescription. The late effects of the surgical load following repeated TURBs, on the other hand, notably compromise the bladder function in our study, hypothesizing a more long-lasting impact on the HRQOL in NMIBC patients.

The management and treatment of NMIBC are fortunately undergoing exciting development, and one of the initiatives that appear capable of reducing the number of TURBs without compromising oncological treatment is chemoresection [35,36]. Long-term results demonstrate a non-different two-year recurrence-free survival in the group receiving chemoresection and the group receiving standard treatment (TURB and possible adjuvant instillation therapy), suggesting that this treatment could have significant potential in NMIBC management, potentially also preserving bladder function. Future studies will elucidate whether the new treatment options for NMIBC leave patients with improved bladder function and, thus, a higher bladder-related quality of life.

## 5. Conclusions

In conclusion, when considering NMIBC patients, the surgical load following repeated TURBs has the highest impact on the bladder function in NMIBC treatment assessed by the use of anticholinergics, β3-agonists, and cystitis-relevant antibiotics. Exposure to adjuvant instillation therapy with BCG or MMC is also associated with the use of anticholinergics, β3-agonists, and cystitis-relevant antibiotics, although a higher number of accumulated instillations appears not directly correlated to an increased use of medication.

## Figures and Tables

**Figure 1 cancers-16-01936-f001:**
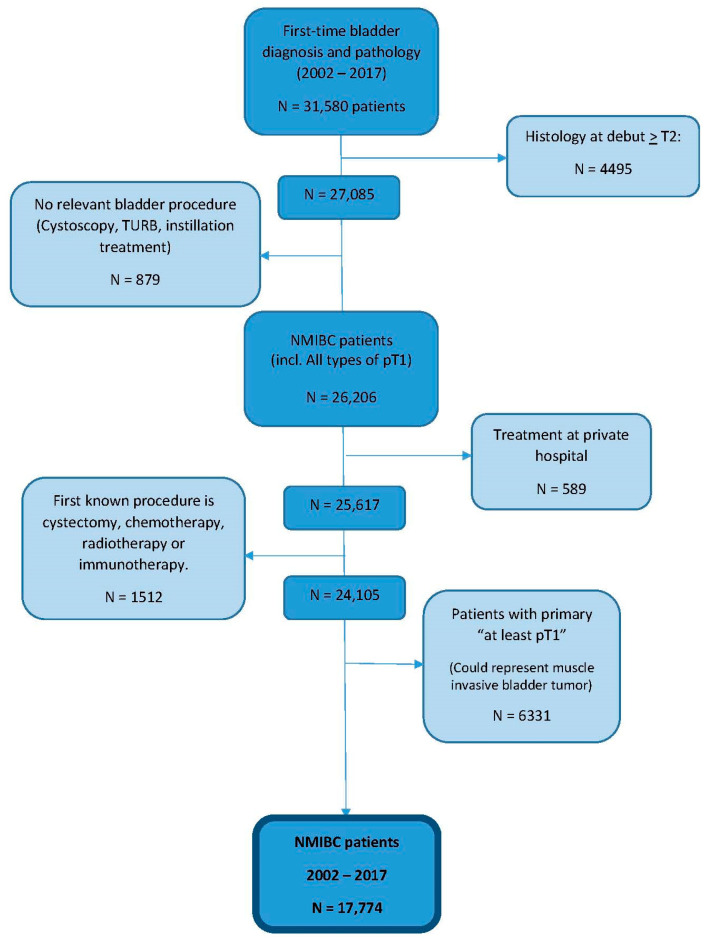
Inclusion flow: N = Sample size; TURB: Transurethral resections of the bladder; NMIBC: Non-muscle invasive bladder cancer.

**Figure 2 cancers-16-01936-f002:**
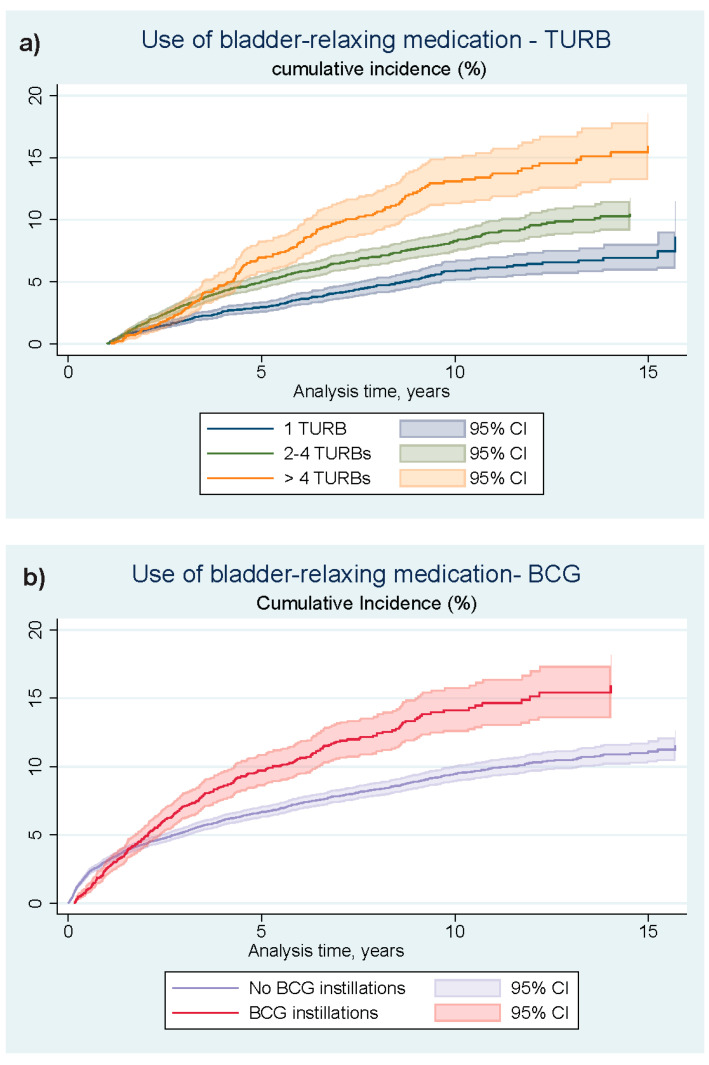
Unadjusted estimates of the cumulative incidences of bladder-relaxing medication use, depending on the exposure of TURB-load within the first five years from NMIBC diagnosis (**a**) or exposure to adjuvant BCG bladder instillation treatment within the first five years from NMIBC diagnosis (**b**). TURB: Transurethral resections of the bladder; NMIBC: Non-muscle invasive bladder cancer; BCG: Bacillus Calmette-Guerin vaccine; 95% CI: 95% confidence interval (a one-year lag-time was added to Figure 2a) simply because it takes time to reach five or more TURBs.

**Figure 3 cancers-16-01936-f003:**
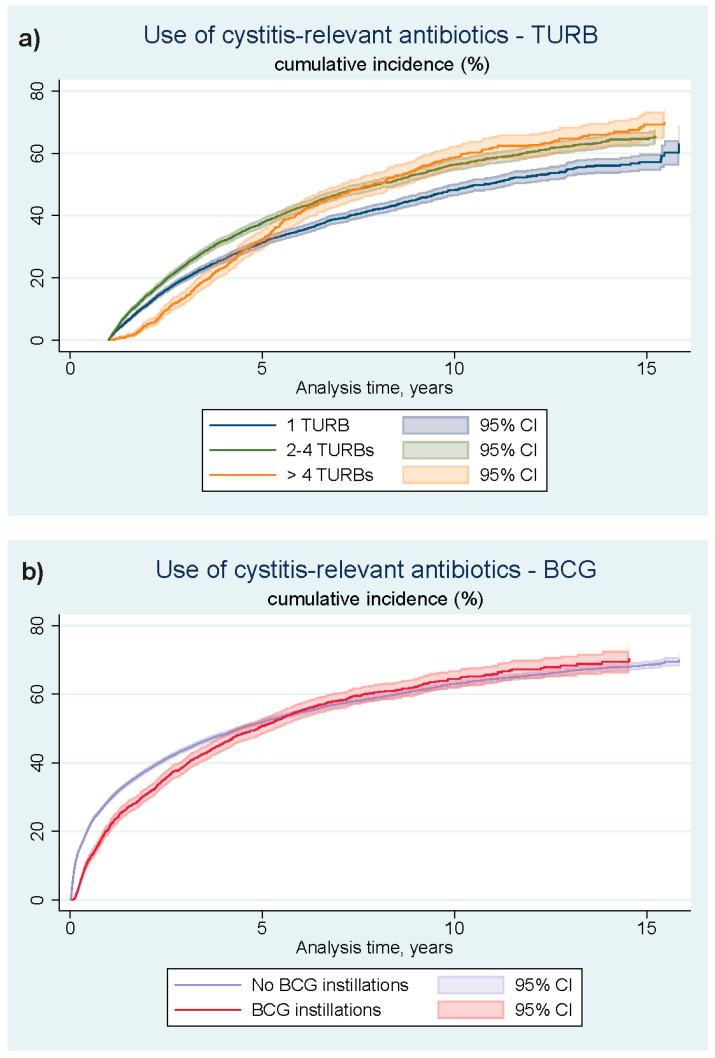
Unadjusted estimates of the cumulative incidences of cystitis-relevant antibiotic use, depending on the exposure of TURB-load within the first five years from NMIBC diagnosis (**a**) or exposure to adjuvant BCG bladder instillation treatment within the first five years from NMIBC diagnosis (**b**). TURB: Transurethral resections of the bladder; NMIBC: Non-muscle invasive bladder cancer; BCG: Bacillus Calmette-Guerin vaccine; 95% CI: 95% confidence interval (a one-year lag-time was added to Figure 2a) simply because it takes time to reach five or more TURBs.

**Table 1 cancers-16-01936-t001:** Overall baseline characteristics.

N	17,774
Sex: female	24.24%
Male	75.76%
Baseline age, median (IQR) *:	70 (63, 77)
Primary histology, freq. (%)	
PUNLMP	886 (4.98%)
pTa LG	8032 (45.19%)
pTa (grade unknown)	785 (4.42%)
pTa HG	2470 (13.90%)
CIS	1085 (6.10%)
pTa, concomitant CIS	570 (3.21%)
pT1a	1485 (8.35%)
pT1 (sub-division unknown)	1491 (8.39%)
pT1b	970 (5.46%)
Any collection of prescriptions before NMIBC ** diagnosis, freq. (%)	
Bladder relaxing agents	1203 (6.77%)
Cystitis-relevant antibiotics	9050 (50.92%)

N = Sample size; * Inter quartile range (IQR); ** Non-muscle invasive bladder cancer (NMIBC).

**Table 2 cancers-16-01936-t002:** Hazard ratios (HR) on collecting two or more prescriptions of bladder-related medication (bladder-relaxing agents or cystitis-relevant antibiotics) depending on exposure (TURB-load, BCG, or MMC).

	Bladder-Relaxing Agents (Anticholinergics and β3-Agonists) *	Cystitis-Relevant Antibiotics *
TURB exposure **	HR [95% conf. interval]	HR [95% conf. interval]
One TURB	1 (reference group)	1 (reference group)
2–4 TURBs	1.98 [1.75; 2.23]	1.77 [1.69; 1.86]
≥five TURBs	4.01 [3.33; 4.83]	2.27 [2.05; 2.51]
Adjuvant BCG treatment ***	HR [95% conf. interval]	HR [95% conf. interval]
No installations	1 (reference group)	1 (reference group)
Yes	1.92 [1.69; 2.18]	1.39 [1.31; 1.48]
Adjuvant MMC treatment ****	HR [95% conf. interval]	HR [95% conf. interval]
No installations	1 (reference group)	1 (reference group)
Yes	2.26 [1.73; 2.95]	1.36 [1.17; 1.59]
Multivariate analysis *****	HR [95% conf. interval]	HR [95% conf. interval]
TURB exposure		
one TURB	1 (reference group)	1 (reference group)
2–4 TURBs	1.86 [1.64; 2.10]	1.74 [1.66; 1.82]
>five TURBs	3.42 [2.82; 4.15]	2.16 [1.95; 2.39]
Adjuvant BCG treatment:		
No	1 (reference group)	1 (reference group)
yes	1.51 [1.33; 1.73]	1.19 [1.12; 1.27]
Adjuvant MMC treatment:		
No	1 (reference group)	1 (reference group)
yes	1.66 [1.27; 2.18]	1.16 [1.00; 1.35]

* Death and cystectomy as competing risks. ** Number of Transurethral resections of the bladder (TURBs) within the first five years from non-muscle invasive bladder cancer (NMIBC)-diagnosis *** Bacillus Calmette-Guerin vaccine (BCG) instillations within the first five years from NMIBC-diagnosis **** Mitomycin C (MMC) instillations within the first five years from NMIBC-diagnosis ***** Including adjustments for TURB exposure and adjuvant instillation treatment with BCG or MMC.

**Table 3 cancers-16-01936-t003:** Sub-analyses: BCG and MMC-treated patients: Hazard ratios (HR) on collecting two or more prescriptions of bladder-relaxing agents (anticholinergics and β3-agonists) and cystitis-relevant antibiotics * depending on accumulated bladder instillations with BCG or MMC. Landmark in analyses: two years (analyses commence 2 years after NMIBC diagnosis).

More than Six Instillations within the First Two Years from NMIBC-Diagnose	Bladder-Relaxing Medication (Anticholinergics and β3-Agonists) HR [95% Conf. Interval]	Cystitis-Relevant Antibiotics HR [95% Conf. Interval]
BCG N = 3365		
No	1 (reference group)	1 (reference group)
Yes	1.34 [0.97; 1.86]	0.80 [0.63; 1.02]
MMC N = 521		
No	1 (reference group)	1 (reference group)
yes	0.52 [0.21; 1.30]	0.95 [0.54; 1.66]

* Death and cystectomy as competing risks. N = sample size; NMIBC: Non-muscle invasive bladder cancer; BCG: Bacillus Calmette-Guerin vaccine; MMC: Mitomycin C.

## Data Availability

Due to Danish law, the datasets presented in this article are not available. Linea Blichert-Refsgaard had full access to all the data in the study and takes responsibility for the integrity of the data and the accuracy of the data analysis.

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
