# Peer review of "Use of Bladder-Related Medication in Non-Muscle Invasive Bladder Cancer Patients"

_cancers, 2024, doi:10.3390/cancers16101936_

Round 1

Reviewer 1 Report

Comments and Suggestions for Authors

This is important study with correctly formulated aim and adequate methodological approach and conclusions. Statistical treatment is done well. This study might be useful for practical urologists.

Author Response

Thank you for the kind response.

Reviewer 2 Report

Comments and Suggestions for Authors

Title: Use of bladder-related medication in non-muscle invasive bladder cancer patients.

Authors: Linea Blichert-Refsgaard, Charlotte Graugaard-Jensen, Mette Nørgaard and Jørgen Bjerggaard Jensen

Affiliation: Department of Urology and Clinical Epidemiology, Aarhus University hospital, Palle Juul-Jensens Boulevard 35, 8200 Aarhus N, 5 Denmark

Authors Conflicts of Interest (COI): None declared by authors.

Reviewer COI: None

The authors report on a retrospective (2002-2017), multicenter, nationwide registry study on the treatment of non-muscle invasive bladder cancer (NMIBC) and related changes in bladder function using bladder-related medication as a surrogate marker of compromised bladder function. They investigated whether the load of repeated surgeries and the exposure to adjuvant instillation influenced bladder function. In their study of 17,774 patients, they found that patients subjected to > 5 TURBTs had higher risk of using bladder-relaxing medication (anticholinergic agents and β3 agonists) than patients exposed to 1 TURBT, and higher risk of cystitis (antibiotics). Adjuvant mitomycin-C and BCG-exposed patients had higher risk of bladder-relaxing medication-use compared to non-exposed, and higher risk of cystitis. They concluded that repeat TURBTs had the highest impact on bladder function and that TURBTs should be minimized in treating NMIBC. This is something that could be potentially achieved with newer therapies including immunotherapies and MMC gel therapy, although not discussed.

Minor concerns: Syntax, spelling, and typo errors. Possibly English as a second language.

Major concerns: This is not a very novel observation and does not contribute significantly to the literature. There is no hypothesis stated. There is no control group (cohort) to show that it was the treatment of bladder cancer that led to the reported observations (In my opinion a fatal flaw). There appears to be a limited understanding of the use of intravesical therapy for NMIBC and risk stratification groups. The two-year lag time biases the study which the authors acknowledge in their report of limitations. The subgroup analysis could have dealt with this by removing those that died before 5 years of follow-up or patients that went on to cystectomy. It would have been relatively easy for the authors to have looked at compliance with intravesical therapy in their registry study, especially if they had risk stratified their patient as low, intermediate, and high risk NMIBC. Some conclusions not supported by their data need to be removed.

Line by line review:

Page 2, line 45; “NMIBC is characterized by a high risk of recurrence and a risk of progression to muscle-invasive bladder cancer; clinical studies have shown an estimated five-year risk of recurrence of NMIBC of up to 78%, and a five-year risk of progression of up to 45%.” Suggest that this can be further risk stratified so as not to over-state the risk (i.e. low, intermediate, and high risk NMIBC).

Page 2, line 51; “….. bacillus Calmette-Guerin vaccine (BCG) is recommended in e.g. European as well as Danish guidelines [4, 5].” Syntax- suggest ‘“….. bacillus Calmette-Guerin vaccine (BCG) is recommended in European Association of Urology (EAU) guidelines, as well as the Danish guidelines [4, 5].’

Page 3, line 131; “Patients with suspension of muscle invasive bladder cancer (TNM >T2 and classification of “at least pT1”) at the time of initial bladder cancer diagnosis, and patients undergoing cystectomy upfront, were excluded.” Typo/spelling error- suggest ‘Patients with suspicion of muscle invasive bladder cancer (TNM >T2 and classification of “at least pT1”) at the time of initial bladder cancer diagnosis, and patients undergoing cystectomy upfront, were excluded.’

Page 3, line 135; “We defined a TURB as an additional pathological bladder cancer diagnosis registered the DPR.” Typo/syntax error- suggest ‘We defined a TURB as an additional pathological bladder cancer diagnosis registered in the DPR.’

Page 3, line 137; “We defined drug use as a collection of > 2 perceptions of a drug of interest, and no- use as 0-1 prescriptions, applicable to both bladder-relaxing agents …..” Typo/syntax error – suggest ‘We defined drug use as a collection of > 2 prescriptions filled of a drug of interest, and non-use as 0-1 prescriptions filled, applicable to both bladder-relaxing agents ….’

Page 5, line 191, Figure 1 and Table 1; If primary pT1 is excluded for concerns of muscle invasive bladder cancer (MIBC) how come T1a, T1b, and T1x are included in overall baseline characteristics?

Page 5, line 195; “Compared to patients with a minimal burden of TURBs, patients with a severe surgical burden of TURBs within the first five years from NMIBC diagnosis, had a higher risk of collecting bladder-relaxing agents, anticholinergics and β3-agonists, with a HR of 197 4.01 95% confidence interval [3.33; 4.83] (table 2).” How where pre-existing prescriptions handled in the interpretation of the data. Did some of these patients have pre-existing overactive bladders?

Page 9, line 255; “The HR of bladder-relaxing agent-use between patients exposed to more than six BCG instillations within the first two years from NMIBC-diagnosis (high accumulated instillations) and the patients exposed to six or les BCG 255 instillations (low accumulated instillations) was 1.34 [0.97; 1.86].” Typo – less, and definition concerns of “high accumulated instillations.” The normal standard induction course for intravesical BCG is once a week for 6 weeks. If high risk NMIBC then most receive maintenance therapy every of 3 weekly intravesical BCG every 3 months as tolerated -mostly to one year. Almost all patients develop symptoms of cystitis after 3 doses at induction. I am surprised that the HR goes down with increased administration as defined in this manuscript

Page 9, line 257; “The HR of bladder-relaxing agents-use between patients exposed to more than six MMC instillations (high accumulated instillations) and the patients exposed to six or less MMC instillations (low accumulated instillations) was 0.52 [0.21; 1.30].” Definition concerns- Again the standard induction course of MMC is a 6 weekly course followed by maintenance. To prevent recurrence from seeding most institutions use one or 2 dose of MMC post TURBT. These arbitrary definitions of high and low accumulation need to be adjusted and a reasonable justification for selection explained to the readers. Not all institutions use preventative MMC due to logistics. How is this handle in Denmark?

Page 10, line 273; “We also found that use of instillation therapies, within the first five years form NMIBC diagnose, was associated with collection of anticholinergics, β3-agonists, and cystitis-relevant antibiotics.” Typo-from

Page 10, 281; “We expected a more significant association between both MMC and BCG instillations and the collection of bladder-related-medicine prescriptions in general.” Hypothesis is finally given in discussion. Suggest that you state the rationale for the study up front. Please see above comments on definition of low and high exposure and how this may confound analysis.

Page 10, line 284; “…. misclassification bias followed by the fixed-time exposure groups in the sub-analysis, and the fact that we introduced a time lag of two years.” A better explanation for the time lag analysis is needed. I would put forward that most therapies are prescribed acutely for symptoms and changes overtime. This could be presented in the data to better inform the interested reader. How do the investigators know this is not new onset overactive bladder (OAB) based on their data set?

Page 10, line 288; “…… accumulated amounts of instillations and bladder impact, towards the null.” I assume the authors are referring to a null hypothesis although a hypothesis was never truly stated and the purpose of the study.

Page 10, line 294; “….towards the null since the patients with an early, large impact on urinary function …” Same as above, what does the null refer to?

Page 10, line 314; “This corresponds to our clinical experience from Denmark.” This data could have easy been extracted from their method of data collection and should be presented to support their premise as opposed to referencing the EAU and Danish guidelines and generalized observations of multiple jurisdictions.

Page 11, line 320; “… the time-sloth short enough to ensure a significant accumulation of instillations over time…” Typo - Freudian slip – I think you meant ‘… the time-slot short enough to ensure a significant accumulation of instillations over time…’

Page 11, line 326; “…associated-estimate towards the null – as they would never reach instillation number …” Does null here refer to ‘zero’ or is it related to a null-hypothesis?

Page 11, line 330; “As so, collection of cystitis-relevant antibiotics might not be a….” Typo/syntax- suggest ‘As such, collection of cystitis-relevant antibiotics might not be a….’

Page 11, line 332; “Nevertheless, a larger use of antibiotics, preventive or not, all contribute to the accumulated risk of developing antibiotic resistance, and every attempt to minimize antibiotic-use should be coveted.” Data does not support this conclusion in the discussion.

Page 11, line 335; “Our study did not include a control group without NMIBC.” Is this not a fatal flaw of the study? Could one not look at a similar cohort of patients with microscopic hematuria undergoing cystoscopy only in the registries?

Page 11, line 346; “Further limitations are mainly due to design and the limited information in the registers.” Typo- suggest ‘Further limitations are mainly due to design and the limited information in the registries.

Page 12, line 373; “This could support our hypothesis that the side effects of instillation therapy are more acute and reversible and, as so, we did not see the impact as strongly when concentrating the outcome on collection of more than one prescription.” Typo/syntax – Never stated as a hypothesis. Suggest ‘This could support our contention that the side effects of instillation therapy are more acute and reversible and, as such, we did not see the impact as strongly when concentrating on the outcome on collection of more than one prescription.’

Page 12, line 417- reference 5; “DaBlaCa [National guidelines, bladder cancer] (accessed may 5. 2022).” Reference is incomplete without web address. Please check other references for formatting and completeness.

Comments on the Quality of English Language

See above detailed review.

Author Response

Thank you for all the usefull comments - a detailed response is attached

Reviewer 3 Report

Comments and Suggestions for Authors

General comment

The manuscript entitled “Use of bladder-related medication in non-muscle invasive bladder cancer patients” presents a comprehensive analysis of the impact of adjuvant instillation therapy and transurethral resections of the bladder (TURB) on bladder function in patients diagnosed with non-muscle invasive bladder cancer (NMIBC). The study employs robust statistical methods and includes detailed analyses of various exposure groups and outcomes. However, several areas need clarification and improvement to strengthen the manuscript.

-          The manuscript would benefit from clearer delineation of methods and results. Some sections, particularly those discussing sub-analyses and the rationale behind certain methodological choices, are convoluted and could be simplified for better readability.

-          While the manuscript mentions the rationale behind certain methodological choices, such as the introduction of lag times and adjustment for confounding variables, a more explicit justification would strengthen the study's validity.

-          About the role of BCG and the effects on bladder need to be add more reference

-          The limitations of the study are discussed; however, further discussion on potential sources of bias and their implications on the interpretation of results would enhance the manuscript's rigor. Additionally, addressing the limitations associated with the use of registry data, such as incomplete information or potential misclassification, would provide a more comprehensive evaluation of the study's validity.

-          The study acknowledges the lack of a control group without NMIBC, which compromises the external validity of the results. Discussing the potential implications of this limitation on the generalizability of findings would be valuable.

Comments on the Quality of English Language

Minor grammar checks

Author Response

(The authors gave the same response as above.)

Round 2

Reviewer 3 Report

Comments and Suggestions for Authors

The authors improved the manuscript accordingly to previous suggestions. The discussion could be further enriched by checking these papers: 10.3390/diagnostics12030586 and 10.3390/curroncol31020079

Comments on the Quality of English Language

None